# Discriminatory Gleason grade group signatures of prostate cancer: An application of machine learning methods

Mpho Mokoatle[1]☉*, Darlington Mapiye[2]☉, Vukosi Marivate[1,3]☉, Vanessa M. Hayes[3,4]‡, Riana Bornman[4]‡

**1** Department of Computer Science, University of Pretoria, Pretoria, South Africa, **2** AstraZeneca, London, United Kingdom, **3** School of Medical Sciences, The University of Sydney, Sydney, Australia, **4** School of Health Systems and Public Health, University of Pretoria, Pretoria, South Africa

☉ These authors contributed equally to this work.
‡ VMH and RB also contributed equally to this work.
* u19394277@tuks.co.za

**Data Availability Statement:** Data cannot be shared publicly because of ethical reasons since it is human samples. Data are available from the University of Pretoria (contact via

## Abstract

One of the most precise methods to detect prostate cancer is by evaluation of a stained biopsy by a pathologist under a microscope. Regions of the tissue are assessed and graded according to the observed histological pattern. However, this is not only laborious, but also relies on the experience of the pathologist and tends to suffer from the lack of reproducibility of biopsy outcomes across pathologists. As a result, computational approaches are being sought and machine learning has been gaining momentum in the prediction of the Gleason grade group. To date, machine learning literature has addressed this problem by using features from magnetic resonance imaging images, whole slide images, tissue microarrays, gene expression data, and clinical features. However, there is a gap with regards to predicting the Gleason grade group using DNA sequences as the only input source to the machine learning models. In this work, using whole genome sequence data from South African prostate cancer patients, an application of machine learning and biological experiments were combined to understand the challenges that are associated with the prediction of the Gleason grade group. A series of machine learning binary classifiers (XGBoost, LSTM, GRU, LR, RF) were created only relying on DNA sequences input features. All the models were not able to adequately discriminate between the DNA sequences of the studied Gleason grade groups (Gleason grade group 1 and 5). However, the models were further evaluated in the prediction of tumor DNA sequences from matched-normal DNA sequences, given DNA sequences as the only input source. In this new problem, the models performed acceptably better than before with the XGBoost model achieving the highest accuracy of 74 ± 01, F1 score of 79 ± 01, recall of 99 ± 0.0, and precision of 66 ± 0.1.

u19394277@tuks.co.za) for researchers who meet the criteria for access to confidential data.

**Funding:** The work reported herein was made possible through funding by the South African Medical Research Council (SAMRC) through its Division of Research Capacity Development under the Internship Scholarship Program from funding received from the South African National Treasury. The content hereof is the sole responsibility of the authors and does not necessarily represent the official views of the SAMRC or the funders. The award recipient is Mpho Mokoatle. The funders had no role in study design, data collection and analysis, decision to publish, or preparation of the manuscript.

**Competing interests:** The authors have declared that no competing interests exist.

# 1 Introduction

Prostate cancer is the leading male cancer in South Africa and is the second most frequently diagnosed cancer among men globally [1]. As men live longer, there is an increase in the occurrence and mortality of the disease [2]. Except for age, the main risk factor is hereditary. Other factors such as race, high-calorie diet, and exposure to heavy metals have a significant impact on the risk of occurring the disease [3, 4].

When it comes to the diagnosis of prostate cancer, a prostate biopsy procedure is common [5]. This procedure involves the extraction of tissue samples from the prostate by using specialised biopsy needles. It is typically performed by using an ultrasound probe that is placed in the rectum which than produces a real-time image of the prostate. The samples produced from this procedure are then taken to a pathologist for evaluation and grading [6, 7].

The Gleason grade group system is the most reliable method and criterion for selection of therapy. In 2014, the International Society of Urological Pathology (ISUP) [8] released supplementary guidance on an improved prostate cancer grading system called the ISUP-Grade Group. This system is simpler, with just five grades, 1 to 5, to describe the growth of the tumor. Grade 1 refers to the least aggressive growth of the tumor, and grade 5 refers to the most aggressive growth [9].

Due to the difficulty and natural subjectivity of this system, Gleason grading is affected by large discordance rates among pathologists (30-50%) [10–15]. However, grades provided by experts with numerous years of experience are more accurate and precise more than grades provided by pathologists with only a few years of experience [16–19], indicating the need to improve the clinical usefulness of the system by improving grading discordance and accuracy [20].

In this work, the DNA sequences that were sequenced from patients that present with a Gleason grade 1 and 5 are studied. The objective of this work is to find discriminatory features within the DNA sequences, and map them to their correct Gleason grade group using machine learning. Two key cancer genes are investigated: *BRCA 1* and *BRCA 2.* These genes have been key genes of interest in prostate cancer [21]. Studies that interrogated these two genes suggest that men who harbor a disease-associated BRCA 2 allele have an increased predisposition of prostate cancer (2 to 5-fold increased risk). This finding suggests that deleterious mutations in BRCA 2 play a significant role in the susceptibility of prostate cancer [22, 23]. Different from BRCA 2 mutations, mutations in BRCA 1 have been inconsistently correlated with the risk of prostate cancer. Studies that have evaluated prostate cancer risk in men that carry BRCA 1 mutations have reportedly been negligible, but not insignificant [24, 25]. The contributions of this work are summarised as follows:

- this study specifically compares two extremes of the Gleason grade group (Gleason grade group 1 and 5)

- while previous studies have used medical images and clinical features [26–31] as input to their Gleason grade group predictor models, this study explores the challenges that are encountered when blood DNA sequences are used as the only input source to the machine learning models.

This work is divided as follows: first, a literature review will be given that highlights the gap in the prediction of the Gleason grade group in the context of machine learning. Second, the data and description of methods will be discussed. Finally, the results, discussion, and conclusion section will follow.

## 2 Literature survey

Recently, deep learning has emerged as a powerful tool to automate the Gleason Grading system. Deep learning systems make use of multi-faceted neural networks that are able to extract complex features from data. Recent work [26] designed a Gleason score annotator by using a convolutional neural network (MobileNet) on tissue microarrays images. The final output layer of this architecture produced a probability distribution over four possible Gleason classes. A key limitation in this work is that the training, testing, and validation sets were too small, which led to some bias in the predictions produced by the model. A recent study [27] similar to this one also used a convolutional neural network (Inception V3) to develop a Gleason score annotator using whole slide images. In addition to predicting a Gleason pattern, this architecture first provided a probability distribution over an image being benign or malignant. Different from the above work, a study [28] applied a convolutional neural network on multi-parametric magnetic resonance imaging (mpMRI) images of prostate cancer patients to extract deep entropy features. Then, the features extracted from the convolutional neural network were used as input to a Random Forest model for prediction of the Gleason grade group. Even though the training data was too small, the performance measure would have been more reliable if the models were cross-validated. Biopsy images of patients who underwent a prostate biopsy following suspicion of prostate cancer has also been used as input to convolutional neural networks (U-Net and an Inception-v3 Network) for the prediction of the Gleason grade group and cancer detection [29, 30]. To validate the performance of the deep learning system, the predictions from the models were compared with those of pathologists where a high agreement was found between the deep learning systems and the pathologists.

Unlike using convolutional neural networks for the prediction of the Gleason grade group, a study [31] developed a machine learning assisted model that predicts the probability of a patient having a Gleason grade upgrade before treatment. The input used to the machine learning models (Logistic Regression, Random Forest, Support Vector Machine) were clinical features such as age, prostate-specific antigen (PSA) level, and the clinical stage.

Overall, much emphasis has been placed on creating machine learning models that predict the Gleason grade group from medical images and clinical data. To the best of our knowledge, this is the first study that focuses on DNA sequences as the only input source to a Gleason grade group prediction model. This work explores the challenges that are associated with finding discriminatory signatures within the DNA sequences of patients that present with a Gleason grade group of 1 and 5.

## 3 Data description, data representation methods, machine learning algorithms, and sequence similarity

### 3.1 Data description

Patients were recruited and consented according to approval granted from the University of Pretoria Faculty of Health Sciences Research Ethics Committee 43/2010 (South Africa); DNA sequencing was generated under approval granted from the St. Vincent's Hospital Human Research Ethics Committee (HREC) SVH/15/227 in Sydney (Australia), and this study was approved by the Faculty of Engineering, Built Environment & IT (Ethics Reference No: 43/2010; 11 August 2020). The data was fully anonymized before analysis.

The DNA sequences of twelve patients with a histopathological ISUP-GG of 1 (low risk prostate cancer) and 5 (high-risk prostate cancer) were selected for analysis. The DNA sequences were aligned using the BWA-MEM aligner [32] to produce BAM files. The BAM

files were converted to FASTA files using samtools [33] and an in-house python script was used for pre-processing and removing IDs from the blood DNA sequences.

The blood DNA sequences were then truncated into $k$-mers. $k$-mers are defined as all the possible substrings of length $k$ that are contained in a sequence [34]. The classification problem in this work is defined as follows: given a DNA sequence $x$ that consists of $k$-mers of size 63, can a machine learning function $f$ learn the correct mapping from the input $x$ to the outcome variable $y$ (Gleason grade group of 5 or 1):

$$y = f(x) \tag{1}$$

After preprocessing, the data was transformed into the below data structure (Fig 1).

## 3.2 Data representation methods

To vectorize the $k$-mers, the Term Frequency—Inverse Document Frequency (TF-IDF) [35] algorithm were used. TF-IDF is a statistical method that calculates how significant a token or word is to a document in a set of documents. Two matrices are used to calculate the TF-IDF score: term frequency (TF), which is a measure of how many times a token appears in a document and inverse document frequency (IDF), is a measurement of how frequent or rare a token is in the entire document set. Multiplying these two measurements produces a TF-IDF score of each word in the document [36]. The main disadvantage of TF-IDF is that it produces extremely high dimensional vectors [37]. To overcome this, the Principal Component Analysis (PCA) [38] was used as a data reduction technique to transform the high dimensional vectors into 2-dimensional (d) vectors.

The other vectorization method that was used was the Skip-gram method from the *word2-vec* algorithm. This vectorization method was chosen as it has been found to be robust with regards to transforming DNA or genomic data into dense vector representations in preparation for machine learning [39–42]. In the context of this work, the usefulness of the Skip-gram model lies in determining $k$-mers that are important in predicting the surrounding $k$-mers in a DNA sequence. Precisely, given a sequence of training $k$-mers $w_1, w_2, w_3, \ldots, w_T$ the training objective of the Skip-gram model is to maximise the average log probability:

$$\frac{1}{T}\sum_{t=1}^{T}\sum_{-c \le j \le c, j \neq 0} log\, p(w_{t+j}|w_t) \tag{2}$$

where $c$ is the size of the context $k$-mers in the training set.

| phenotype | kmers |
|---|---|
| 1 | [' tactg', 'tactgg', 'actggg', 'ctgggc', 'tggg... |
| 1 | [' ttaaa', 'ttaaaa', 'taaaaa', 'aaaaat', 'aaaa... |
| 0 | [' aatag', 'aataga', 'atagac', 'tagact', 'agac... |
| 1 | [' ttcct', 'ttcctt', 'tccttc', 'ccttct', 'cttc... |
| 0 | [' tgacc', 'tgacct', 'gacctg', 'acctgc', 'cctg... |

**Fig 1. Blood DNA sequences $x$ transformed into $k$-mers with their corresponding Gleason grade group $y$.**

In this work, the Skip-gram $k$-mer tokens were represented by a continuous vector of size 100, and summed up with other vectors of the same sequence to give a single continuous vector that represents the entire sequence.

## 3.3 Machine learning algorithms

After obtaining the 2-d TF-IDF vectors from PCA, they were used as features to several machine learning models: Gradient boosting algorithm: eXtreme Gradient Boosting (XGBoost), Long Short-Term Memory (LSTM), Gated Recurrent Unit (GRU), and Random Forest (RF). XGBoost is an ensemble boosting learning method that makes use of several learners to make predictions. This method is different from other ensemble methods as it builds a sequence of originally weak models into progressively more powerful models, where the errors made by previous models are corrected in subsequent models [43]. The steps involved in the ensemble technique are as follows:

- first, an initial model $F_o$ is initialised to predict the target variable $y$. This model produces a residual error $(y − F_o)$.

- next, an additive learner $h_1$ is fit onto the the residuals from the previous step.

- than, $F_o$ and $h_1$ are summed to produce $f_1$, which is the boosted version of $f_o$. The residual error from $f_1$ will be lower in comparison to the residuals of $f_o$:

$$F_1(x) \Leftarrow F_o + h_1(x) \tag{3}$$

To improve the performance of $f_1$, the residuals of $f_1$ can be modeled to create a new model $f_2$:

$$F_2(x) \Leftarrow F_1(x) + h_2(x) \tag{4}$$

This procedure can be performed for a few iterations $m$ until residual errors have been minimised as much as possible:

$$F_m(x) \Leftarrow F_{m-1}(x) + h_m(x) \tag{5}$$

Instead of fitting the additive learners $hm(x)$ on the residuals, fitting it on the gradient of the loss function makes this process more generic and applicable across all loss functions. Hence, XGBoost uses the gradient descent algorithm to minimise the loss [43, 44].

The LSTM and GRU are variants of Recurrent Neural Networks (RNNs) that regulate information through the network by using several gates. The gates regulate the flow of information by learning which timestamps are important to keep or discard [45].

In an LSTM cell (Fig 2) the sigmoid function called the forget gate is responsible for deciding which information will be discarded from the cell state. This gate takes as input $x_t$ and the previous hidden state $h_{t-1}$, than outputs a value between 0 and 1 for each value in the cell state $C_{t-1}$. If the value is 1, the information from the previous hidden state will be kept and if the value is 0, the information from the previous previous hidden state will be discarded [46–48]:

$$f_t = \sigma(W_f \cdot [h_{t-1}, x_t] + b_f) \tag{6}$$

Next, the input gate $i_t$ has to determine which new information will be added in the cell state. Than, a *tanh* layer will create a vector of new candidate values $\tilde{C}_t$, that will be added to

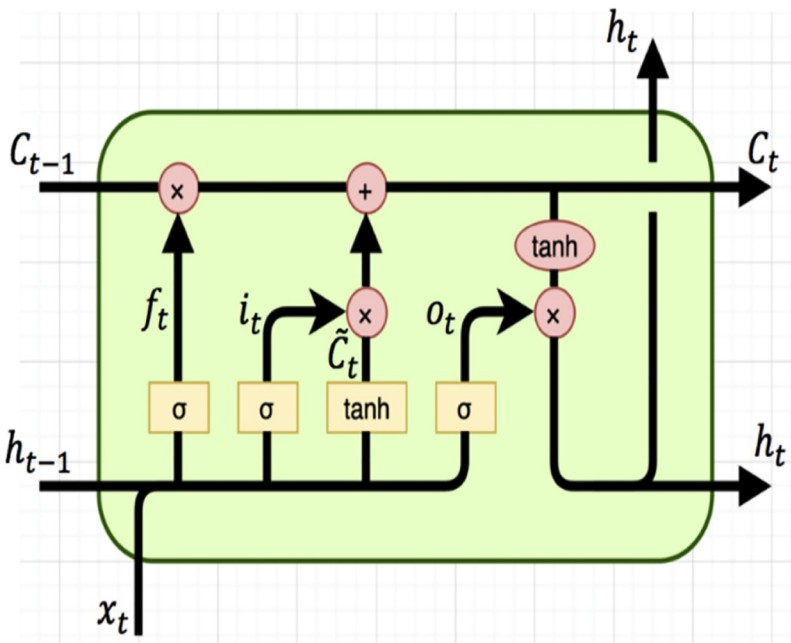

**Fig 2. Architecture of an LSTM unit [59].**

the cell state:

$$i_t = \sigma(W_i \cdot [h_{t-1}, x_t] + b_i) \tag{7}$$

$$\tilde{C} = tanh(W_C \cdot [h_{t-1}, x_t] + b_C) \tag{8}$$

To update the old cell state $C_{t-1}$ into the new cell state $C_t$, the old state is multiplied by $f_t$. Next, $i_t * \tilde{C}_t$ is added, which are the new candidate values:

$$C_t = f_t * C_{t-1} + i_t * \tilde{C}_t \tag{9}$$

Finally, the output that is based on the cell state is given. The cell state is put through a *tanh* function and multiplied by the output of the sigmoid gate: [46–48]

$$o_t = \sigma(W \circ [h_{t-1}, x_t] + b_\circ) \tag{10}$$

$$h_t = o_t * tanh(C_t) \tag{11}$$

A three layer LSTM architecture was selected with a total of 224 hidden units. The output layer consisted of a sigmoid activation function that provides a probability distribution of a sequence either belonging to a patient with a Gleason grade group of 1 or 5. The training dataset was divided over 50 batches and trained over 5 epochs. Dropout rate at 60% was used to control overfitting of the model.

GRUs (Fig 3) are similar to LSTMs in that they both use gates to regulate the flow of information. GRUs are faster to train than LSTMs, and also have have a simpler architecture [49–51].

Inside a GRU cell, at each timestamp *t*, the cell takes an input $X_t$ and the hidden state $h_{t-1}$ from the previous timestamp. Next, the cell will output a new hidden state $h_t$ which will be fed

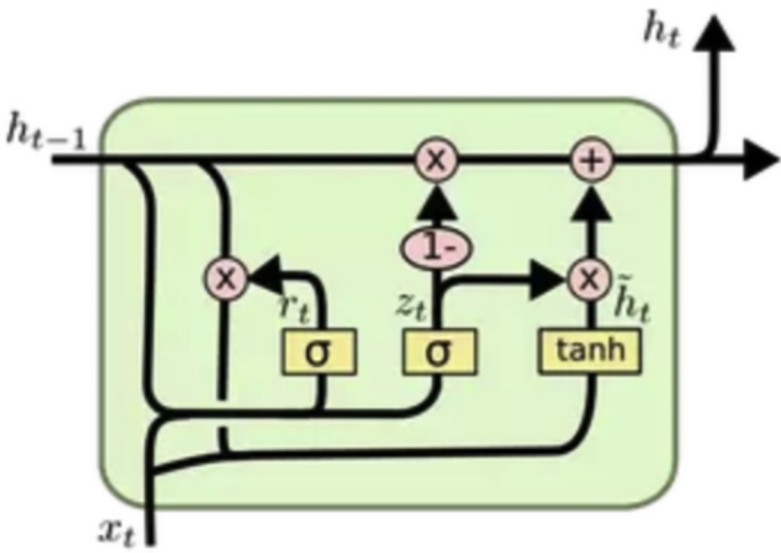

**Fig 3. Architecture of a GRU unit [59].**

as input to the next timestamp. Unlike the LSTM that has three gates, the GRU has two gates: the update gate and the reset gate. The reset gate $r_t$ is in charge of the short-term memory of the network. It is responsible for deciding which timestamps to discard [49–51]:

$$r_t = \sigma(X_t * U_r + H_{t-1} * W_r) \tag{12}$$

$r_t$ will output a value between 0 and 1 due to the sigmoid function. As previously mentioned, if the output value is equal to 1, this means that the timestamps from the previous hidden state $h_{t-1}$ will be kept. And if the output value is 0, the timestamps from the previous hidden state $h_{t-1}$ will be discarded [49–51].

To generate the hidden state $\hat{H}_t$ of a GRU cell, a two-step process is followed. First, a candidate hidden state $\hat{H}_t$ needs to be generated:

$$\hat{H}_t = tanh(X_t * U_g + (r_t \circ H_{t-1}) * W_g) \tag{13}$$

the input $X_t$ and the hidden state from the previous timestamp $H_{t-1}$ are multiplied by the output of the reset gate $r_t$. Next, this is passed to a *tanh* function which outputs the candidates hidden state $\hat{H}_t$. The usefulness of this equation is important in showing how the value of the reset gate is used to control how much influence the previous hidden state can have on the candidate state [49–51].

Similarly, the GRU cell also has an update gate which is responsible for determining how much past information needs to be kept:

$$u_t = \sigma(X_t * U_u + H_{t-1} * W_u) \tag{14}$$

This equation is similar to the one used by the reset gate, the only key differences are the new weight matrices $U_u$ and $W_u$ [49–51].

The GRU models were configured with a stack of four hidden layers and a total of 240 hidden units. The output layer was also a dense layer with a sigmoid activation function, and the model was trained over 5 epochs with the training set divided over 50 batches. Dropout (at

60%) was also used to control overfitting. All the machine learning models were validated via a Repeated $k$-fold cross validation (cv) (cv = 5, runs = 5). The experiments in this work were conducted on a NVIDIA Tesla P100 GPU virtual machine with 100 GB of memory.

RF was also used to find discriminatory signatures between Gleason grade group 1 and 5 blood DNA sequences. In RF, several decision trees are created simultaneously. In the final prediction, the multiple decision trees are merged in order to determine the final answer, which will be the average of all the decision trees [52]. To decide how the nodes of the decision trees would branch, the default Gini index was used:

$$Gini = 1 - \sum_{i=1}^{c} (p_i)^2 \tag{15}$$

Where $p_i$ is the relative frequency and $c$ represents the number of classes. This equation makes use of the class and probability to determine the Gini of each branch on a node [52].

Another binary machine learning model that was used was the Logistic Regression (LR). The Skip-gram $k$-mer features were used as input to a Logistic Regression (LR) model. A logistic regression model is a machine learning model that uses a decision boundary to separate a set of data points into their distinct classes. A logistic regression is comparable to linear regression, the key difference between them is that logistic regression is used when the target variable is categorical, while linear regression is used when the target variable is continuous. In this study, the target variable is categorical (1 = Gleason grade group of 5, 0 = Gleason grade group of 1). Logistic regression uses a Sigmoid function to convert the probability values $z$ to be in the range between 0 and 1:

$$S(z) = \frac{1}{1 + \epsilon^z} \tag{16}$$

This function transforms $-\infty$, 0 and $+\infty$ to 0, 0.5, and 1 respectively. If the probability value $z$ for a data point is close to $+\infty$, this is an indication that the data point is above the decision boundary, hence it will belong to the positive class. In contrast, If the the probability value $z$ for a data point is close to $-\infty$, it means that the data point is below the decision boundary, meaning it belongs to the negative class. If the data point is predicted to be on the decision boundary, the value of $z$ is 0, and the Sigmoid function will transform it to 0.5, meaning that it has a 50% probability of belonging to the positive class [53, 54].

## 3.4 Sequence similarity

Multicollinearity is a problem in machine learning where two or more predictor variables are highly correlated with each other [55]. This presents a problem because the individual effects of the predictor variable on the target variable would not be distinguishable. One of the methods that is applied to deal with multicollinearity in machine learning is to remove the collinear variables. In the context of this work, removing collinear $k$-mers would result in a completely new set of DNA sequences since the sequences would have to be truncated either in the beginning, middle, or at the end. In the context of this work, multicollinearity can also be equated to sequence similarity in genomics. Sequence similarity is an important concept in genomics that refers to the degree of similarity between sequences [56]. This is often indicated as a percentage of identical bases over a given length of the alignment. The Basic Local Alignment Search Tool (BLAST) was used to evaluate the similarity between blood DNA sequences [57]. When a sequence similarity test is performed between a pair of sequences, several attributes are returned such as the *E value, query cover*, and *percent identity*. In this work, only the *percent identity* is reported. The percent identity refers to how similar the query sequence is to the

subject sequence. Specifically, it describes the number of bases that are identical in the sequences. A significant match is 100% [58].

A figure (Fig 4) has been generated to provided an overview of all the methods that were used in this work.

## 4 Results and discussion

### 4.1 Sequence similarity results and TF-IDF Visualizations

For both *BRCA 1* and *BRCA 2*, the results (Tables 1 and 2) illustrate that most sequences are highly similar with a percent identity of 90-100%. The lowest percent identity across the sequences is 70-80%, which is still too high. This indicates that blood DNA sequences that are derived from patients that present with Gleason grade group of 5 are not that very different from patients that present with a Gleason grade group of 1. There might exist a small region of dissimilarity, however, at this stage, the number of sequences available for this experiment are inadequate to capture the region of dissimilarity. It is probable that hundreds of thousands of DNA sequences are required to capture this region.

Next, the impact of this high similarity is investigated in the machine learning models to determine if discriminatory signatures (region of dissimilarity) within the DNA sequences can be detected and mapped to their correct Gleason grade group.

To ensure that the machine learning models are trained on distinct sequences, highly similar sequences were removed using BLAST. Before the removal of highly similar sequences, the total number of blood DNA sequences from the *BRCA 1* gene were 235 711. For *BRCA 2*, the total number of the sequences were 243 822. After the removal of highly similar sequences, the table (Table 3) shows the new data distribution and the total number of sequences in each class. Blood DNA sequences that shared more than 25 bases of homology were considered as similar and were thus removed.

In keeping with the high sequence similarity observation amongst the blood DNA sequences as shown above, the TF-IDF visualisation of the *k*-mers (Figs 5 and 6), also show that there is a great overlap between the *k*-mer features of the two Gleason grade groups as no separable clusters were detected.

### 4.2 Machine learning results

The RF model achieved the highest accuracy as shown (Table 4). However, the recall was too high. This is an indication that the majority of the DNA sequences were predicted as positive

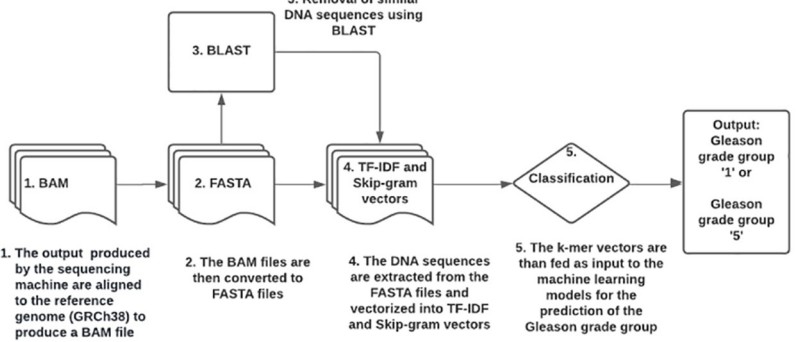

**Fig 4. This figure represents the summary of all the methods that were executed in this work.**

**Table 1. Sequence similarity within a Gleason grade group of 5 and 1 for *BRCA 1* blood DNA sequences.**

|  | Grouped by percentage of identical matches | Total no. of local alignments |
|---|---|---|
| **Gleason grade group 5** | 90-100 | 7170891 |
|  | 80-90 | 3685304 |
|  | 70-80 | 62500 |
| **Gleason grade group 1** | 90-100 | 7270628 |
|  | 80-90 | 3732281 |
|  | 70-80 | 56560 |

**Table 2. Sequence similarity within a Gleason grade group of 5 and 1 for *BRCA 2* blood DNA sequences.**

|  | Grouped by percentage of identical matches | Total no. of local alignments |
|---|---|---|
| **Gleason grade group 5** | 90-100 | 6256450 |
|  | 80-90 | 910123 |
|  | 70-80 | 17970 |
| **Gleason grade group 1** | 90-100 | 6510144 |
|  | 80-90 | 932427 |
|  | 70-80 | 16167 |

**Table 3. Data count and distribution of classes after the removal of highly similar DNA sequences.**

|  | Gleason grade group 5 | Gleason grade group 1 |
|---|---|---|
| **BRCA 1** | 3111 ∼ 58% | 2210 ∼ 42% |
| **BRCA 2** | 3108 ∼ 62% | 1941 ∼ 38% |

(Gleason grade group 5), with very few true negatives (Fig 7). This trend was also observed with the other models as well, which is an indication that not enough learning was achieved.

Considering the results of the *BRCA 2* gene (Table 5), the LR and GRU models achieved the highest accuracy while having the highest recalls indicating that a large number of sequences were predicted as positive. The confusion matrix of the GRU model is shown (Fig 8).

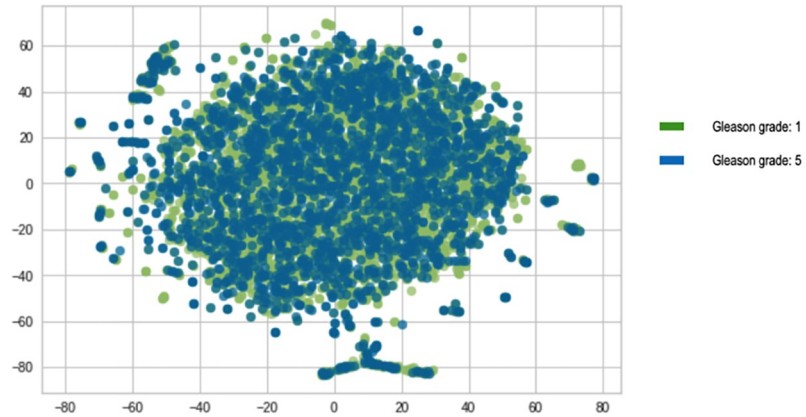

**Fig 5. Visualisation of TF-IDF *kmers* for *BRCA 1*.**

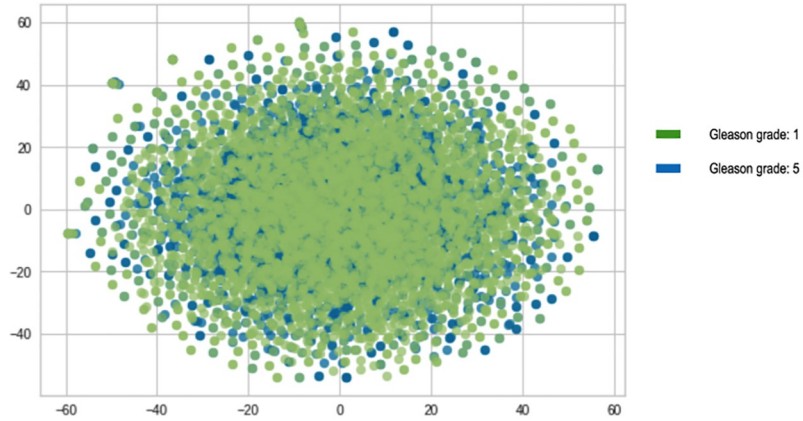

**Fig 6. Visualisation of TF-IDF *kmers* for *BRCA 2 kmers*.**

**Table 4. This table shows the results of the machine learning models using data from the *BRCA 1* gene.**

|                   | Acc (%)    | F1 (%)     | Recall (%)  | Precision (%) |
|-------------------|------------|------------|-------------|---------------|
| **XGBoost**       | 57 ± 1.6   | 69 ± 1.3   | 85 ± 2.0    | 58 ± 1.8      |
| **LSTM**          | 58 ± 1.5   | 74 ± 1.3   | 100 ± 0.0   | 58 ± 1.5      |
| **GRU**           | 58 ± 1.1   | 74 ± 0.9   | 100 ± 0.0   | 58 ± 1.1      |
| **LR**            | 58 ± 1.7   | 73 ± 1.3   | 98 ± 0.7    | 58 ± 1.6      |
| **Random Forest** | 59 ± 1.7   | 74 ± 1.4   | 98 ± 0.8    | 59 ± 1.7      |

While some of machine learning models achieved just above average performance, they all seemed to classify most blood DNA sequences as positive (Gleason grade group 5), which suggests that no discriminatory signatures were discovered within the blood DNA sequences of patients that present with a Gleason grade group of 5 and Gleason grade group of 1. This finding further stipulates that are still a lot of opportunities for improvement with regards to designing more robust data representation methods and machine learning classifiers that

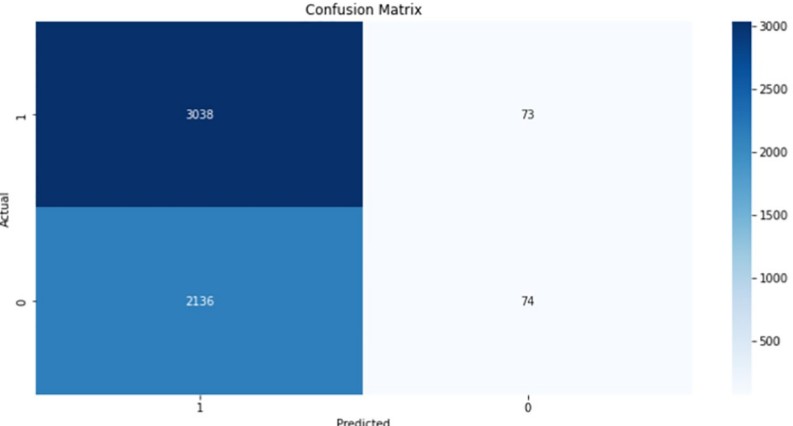

**Fig 7. Confusion matrix of the Random Forest model for *BRCA 1*.**

**Table 5. This table shows the results of the machine learning models using data from the *BRCA 2* gene.**

|  | Acc (%) | F1 (%) | Recall (%) | Precision (%) |
|---|---|---|---|---|
| **LSTM** | 58 ± 1.5 | 73 ± 1.3 | 100 ± 0 | 58 ± 1.6 |
| **XGBoost** | 61 ± 1.3 | 74 ± 1 | 93 ± 1.3 | 62 ± 1.4 |
| **Random Forest** | 61 ± 0.1 | 75 ± 0.8 | 99 ± 0.6 | 61 ± 1.1 |
| **LR** | 62 ± 1.3 | 76 ± 0.1 | 99 ± 0.2 | 62 ± 1.3 |
| **GRU** | 62 ± 1.2 | 77 ± 0.9 | 100 ± 0 | 62 ± 1.2 |

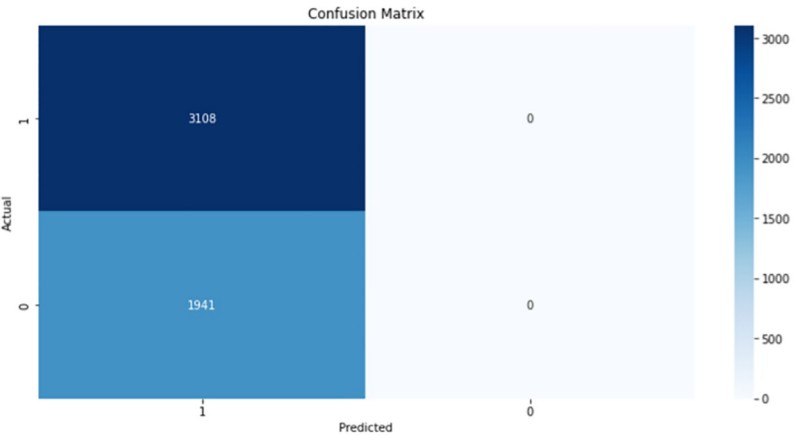

**Fig 8. Confusion matrix of the GRU model for *BRCA 2*.**

are adequately sensitive to detect discriminatory Gleason grade groups signatures in DNA sequences.

## 4.3 Prediction of tumor DNA sequences

Having observed that the above machine learning models were not able to adequately find discriminatory signatures in the DNA sequences of the two Gleason grade groups, a new classification question was formulated: *Given tumor and matched-normal DNA sequences, can the models predict tumor DNA sequences?*. This new problem was formulated to further assess the usefulness of the machine learning models and determine if other classification problems can be learned using DNA sequences as the only input source to the models. In addition, a bigger dataset was used that contained 304 450 tumor DNA sequences and 305 214 matched-normal DNA sequences from the *APC* gene of colorectal cancer patients.

The three machine learning models (LR, RF, and XGBoost) were evaluated to establish if they can distinguish tumor DNA sequences from normal DNA sequences. The results (Table 6) show an overall improvement in the performance of the models compared to the results seen in the previous section of the prediction of the Gleason grade group. In the

**Table 6. This table shows the results of the machine learning models using data from the *APC* gene.**

|  | Acc (%) | F1 (%) | Recall (%) | Precision (%) |
|---|---|---|---|---|
| **LR** | 65 ± 0.1 | 67 ± 0.1 | 71 ± 0.1 | 63 ± 0.1 |
| **Random Forest** | 71 ± 0.1 | 75 ± 0.3 | 87 ± 0.3 | 66 ± 0.3 |
| **XGBoost** | 74 ± 0.1 | 79 ± 0.1 | 99 ± 0.0 | 66 ± 0.1 |

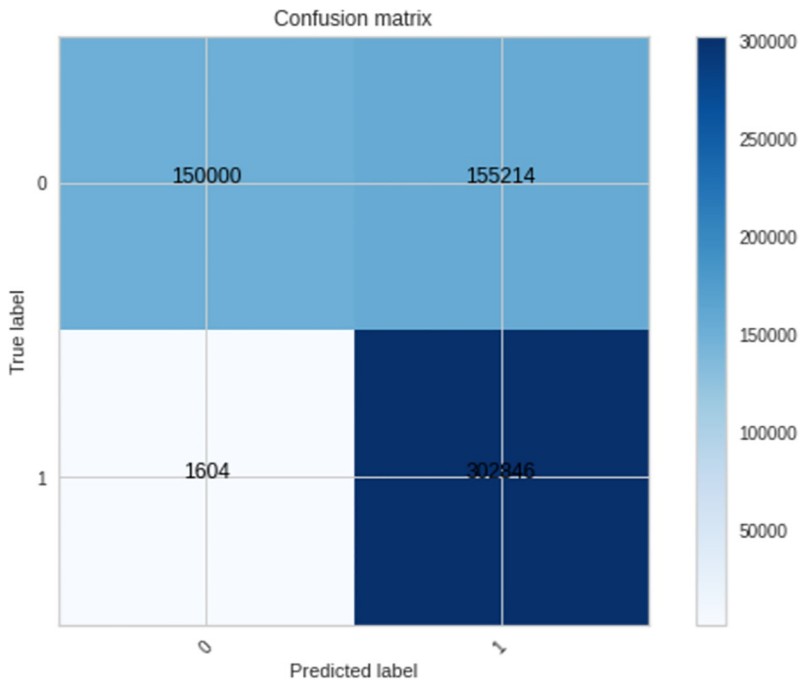

**Fig 9. Confusion matrix of the XGBoost model for the *APC* gene.**

previous section, the models struggled to predict the Gleason grade group given DNA sequences and in this section of results, although there is plenty of room for improvement; the models were able to satisfactorily separate tumor DNA sequences from matched-normal DNA sequences. The confusion matrix of the highest performing model (XGBoost) is shown (Fig 9).

The main limitations of this work include the use of a small sample size, particularly the *BRCA 1* and *BRCA 2* DNA sequences. For this reason, the machine learning models were not able to competently distinguish Gleason grade group of 5 DNA sequences from Gleason grade group of 1 DNA sequences. The other limitation in this work include the lack of sufficient prior research on this topic, particularly research that has used DNA sequences as the only input source to machine learning or deep learning classifiers in the prediction of the Gleason grade group problem. Subsequently, it was difficult to benchmark the results of this work with those in the literature.

## 5 Conclusion

The goal of this work was to apply machine learning algorithms in the prediction of the Gleason grade group in blood DNA sequences of high-risk and low-risk prostate cancer patients. The machine learning models were not able to sufficiently discriminate between Gleason grade group of 5 DNA sequences from Gleason grade group of 1 DNA sequences. The reasons for this occurred as a result of having a large number of sequences that share a substantial amount of sequence homology. Even though this was circumvented by removing highly similar sequences, it was still not sufficient as the machine learning classifiers still produced a high number of false positives and a negligible amount of true negatives. Since the machine learning models were not able to discriminate between the DNA sequences of the two Gleason grade groups, they were further evaluated to determine their usefulness in the prediction of tumor

DNA sequences from matched-normal DNA sequences. In this new problem, the models performed acceptably better than before.

The future work involves the design of better data representation techniques that are sensitive enough to discover discriminatory signatures in small sample sizes of DNA sequences. These techniques should be generic in that they should not only be sensitive towards Gleason grade groups, but should extend to other prediction problems that are important in machine learning and cancer research.

## Acknowledgments

The authors would like to thank the patients for their valued contributions, as well the Southern African Prostate Cancer Study (SAPCS), for providing the data used in the study. The authors would also like to thank fellow colleagues that reviewed this manuscript.

## Author Contributions

**Conceptualization:** Mpho Mokoatle, Darlington Mapiye.

**Data curation:** Mpho Mokoatle, Vanessa M. Hayes, Riana Bornman.

**Formal analysis:** Mpho Mokoatle, Darlington Mapiye.

**Investigation:** Mpho Mokoatle.

**Methodology:** Mpho Mokoatle, Darlington Mapiye.

**Resources:** Mpho Mokoatle.

**Supervision:** Vukosi Marivate.

**Validation:** Mpho Mokoatle, Vanessa M. Hayes, Riana Bornman.

**Visualization:** Mpho Mokoatle.

**Writing – original draft:** Mpho Mokoatle.

**Writing – review & editing:** Mpho Mokoatle, Vukosi Marivate.

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
