## [Decision Letter · Decision Letter 0]

27 Jan 2022

PONE-D-21-39391Discriminatory Gleason grade group signatures of prostate cancer: An application of  machine learning methodsPLOS ONE

Dear Dr. Mokoatle,

Thank you for submitting your manuscript to PLOS ONE. After careful consideration, we feel that it has merit but does not fully meet PLOS ONE’s publication criteria as it currently stands. Therefore, we invite you to submit a revised version of the manuscript that addresses the points raised during the review process.

We look forward to receiving your revised manuscript.

Kind regards,

Sathishkumar V E

Academic Editor

PLOS ONE

Journal Requirements:

"No. The funders had no role in study design, data collection and analysis, decision to publish, or preparation of the manuscript."

"No. The authors have declared that no competing interests exist."

Reviewers' comments:

Reviewer's Responses to Questions

**Comments to the Author**

1. Is the manuscript technically sound, and do the data support the conclusions?

Reviewer #1: Partly

Reviewer #2: Yes

2. Has the statistical analysis been performed appropriately and rigorously? 

Reviewer #1: Yes

Reviewer #2: No

3. Have the authors made all data underlying the findings in their manuscript fully available?

Reviewer #1: Yes

Reviewer #2: No

4. Is the manuscript presented in an intelligible fashion and written in standard English?

Reviewer #1: Yes

Reviewer #2: Yes

5. Review Comments to the Author

Reviewer #1: This article focus on Gleason grade group in blood DNA sequences of high-risk and low-risk prostate cancer prediction using machine learning algorithms. This article primarily focus on similarity index and robust data representation methods where majority of the existing algorithms are lack. This article is partially novel whereas the contributions are very limited for a journal article. The decision on this manuscript may be considered after addressing the below:

1. There are several machine learning algorithms in the literaute, and it is confusing that which machine learning algorithms considered by the authors. It is recommended to mention the accurate machine learning algorithm used in the paper, instead of generalizing the machine learning.

2. Recommended to summarize the list of contributions of the article in the introduction.

3. The literature of the article is very poor. It is recommended to consider the recent literature on the specified topic. There are several articles published recently on this topic, and consider the papers published in last three years.

4. It is recommending the authors to summarize the details about each studied paper and and list the limitations. Which limitations of the existing paper are addressed in the proposed work to be mentioned.

5. Section 3 is too small. It may be written as a subsection under section 4. Recommended to provide the citations for the datasets considered for this work.

6. Section 4 name, (Methods) may be replaced with the actual name of the proposed work.

7. The name of subsection 4.3 must be changed.

8. The proposed method is not clear. Recommended to explain the proposed work through illustrative example.

9. the problem formulation and objective is not clear. It is recommended to provide through theoretical discussion.

10. Discuss about the computational complexity of the proposed model.

11. It is not clear that how the machine learning algorithm address the challenges discussed on the problem.

12. The limitations of the proposed work must be discussed.

13. The experimental results are not enough to justify the performance of the proposed work. Recommended to consider the more metrics with different datasets to justify the performance of the proposed work.

14. Add the future scope of the work in the conclusion.

Reviewer #2: 1. Abstract must conclude the findings with quantitative results

2. Experiment analysis with other machine learning methods is required

3. Architecture diagram depicting LSTM and GRU need to be included

4. Statistical validation of the results is required

6. PLOS authors have the option to publish the peer review history of their article (what does this mean?). If published, this will include your full peer review and any attached files.

Reviewer #1: No

Reviewer #2: No

---

## [Author Response · Author response to Decision Letter 0]

6 Apr 2022

Dear Editors

We would like to thank and appreciate your generous time and comments on reviewing the manuscript and have revised it to address your concerns.

We believe that the manuscript is now in a suitable position for publication.

Reviewer # 1:

Comment: 1. There are several machine learning algorithms in the literaute, and it is confusing that which machine learning algorithms considered by the authors. It is recommended to mention the accurate machine learning algorithm used in the paper, instead of generalizing the machine learning.

Response:

Initially, the machine learning models where briefly described. In the revised version, all the machine learning models, and how they work are described (see section 3.3).

Comment 2: Recommended to summarize the list of contributions of the article in the introduction.

Response:

Agreed. The key contributions are now summarised in bullet form at the end of the introduction section.

Comment 3:The literature of the article is very poor. It is recommended to consider the recent literature on the specified topic. There are several articles published recently on this topic, and consider the papers published in last three years.

Response:

The literature review has been improved and includes recent literature [2018-2020].

Comment 4: It is recommending the authors to summarize the details about each studied paper and and list the limitations. Which limitations of the existing paper are addressed in the proposed work to be mentioned.

Response:

The studied papers have been summarised and the limitations have been stated where necessary. The relationship between the literature and the problem formulated in this paper has also been stated at the end of the literature review section.

Comment 5: Section 3 is too small. It may be written as a subsection under section 4. Recommended to provide the citations for the datasets considered for this work.

Response:

Done. Section 3 has been moved under section 4. At this point, it is not possible to provide the citations of the datasets considered in this work as they have not been published or made freely available to the public. However, access to the data can be granted from the authors; provided that all ethical procedures are followed.

Comment 6: Section 4 name, (Methods) may be replaced with the actual name of the proposed work.

Response:

Section 4 (Now section 3) includes all the data preparation steps and machine learning algorithms used in this paper. It cannot be renamed to the actual name of the proposed work as no new algorithm has been proposed by this work. All the techniques used in this work already exist in the literature. The application thereof is the one that is unique.

Comment 7: The name of subsection 4.3 must be changed.

Response:

Subsection 4.3 now renamed to subsection 3.3 cannot be changed as it describes all machine learning learning algorithm that was used in this work.

Comment 8: The proposed method is not clear. Recommended to explain the proposed work through illustrative example.

Response:

Figure 4 has been generated to provide a more detailed overview of all the methods followed in the paper from the preprocessing steps to the machine learning algorithms.

Comment 9: the problem formulation and objective is not clear. It is recommended to provide through theoretical discussion.

Response:

An attempt of clarifying the problem and objective has been made by highlighting the gaps in the literature at the end of the literature review; and also summarising the contributions of this work at the end of the introduction section.

Comment 10: Discuss about the computational complexity of the proposed model.

Response:

The models used in this work and how they work have been described in the section 3.

Comment 11: It is not clear that how the machine learning algorithm address the challenges discussed on the problem.

Response:

The machine learning models were used to find discriminatory features within the DNA sequences, and try to map them to their correct phenotype (Gleason grade group of 1 or 5). To do this, binary classification models were created.

Comment 12: The limitations of the proposed work must be discussed.

Response:

The limitations of this work are summarised at the end of the Results and discussion section (section 4).

Comment 13: The experimental results are not enough to justify the performance of the proposed work. Recommended to consider the more metrics with different datasets to justify the performance of the proposed work.

Response:

More experiments have been done on a new and bigger dataset (APC gene), and an additional machine learning model (Random forest) was added to the experiments.

Comment 14: Add the future scope of the work in the conclusion.

Response:

Done. The future work has been outlined towards the end of the conclusion section.

Reviewer # 2:

Comment 1: Abstract must conclude the findings with quantitative results

Response:

Done. The abstract now includes the summary of the results.

Comment 2: Experiment analysis with other machine learning methods is required

Response:

An additional machine learning algorithm was used (Random forest). Also, the models were ran on a bigger dataset (APC gene).

Comment 3: Architecture diagram depicting LSTM and GRU need to be included

Response:

Done. Figure 2 and 3 depicts an LSTM and GRU unit.

Comment 4: Statistical validation of the results is required

Response:

The machine learning models were rerun and validated with the Repeated Kfold cross-validation. This validation method returned the average accuracy, F1 score, recall, and precision and average standard deviation on the held-out data from the folds.

---

## [Decision Letter · Decision Letter 1]

14 Apr 2022

Discriminatory Gleason grade group signatures of prostate cancer: An application of  machine learning methods

PONE-D-21-39391R1

Dear Dr. Mokoatle,

We’re pleased to inform you that your manuscript has been judged scientifically suitable for publication and will be formally accepted for publication once it meets all outstanding technical requirements.

Kind regards,

Sathishkumar V E

Academic Editor

PLOS ONE

Additional Editor Comments (optional):

Reviewers' comments:

Reviewer's Responses to Questions

**Comments to the Author**

1. If the authors have adequately addressed your comments raised in a previous round of review and you feel that this manuscript is now acceptable for publication, you may indicate that here to bypass the “Comments to the Author” section, enter your conflict of interest statement in the “Confidential to Editor” section, and submit your "Accept" recommendation.

Reviewer #1: All comments have been addressed

2. Is the manuscript technically sound, and do the data support the conclusions?

Reviewer #1: Yes

3. Has the statistical analysis been performed appropriately and rigorously? 

Reviewer #1: Yes

4. Have the authors made all data underlying the findings in their manuscript fully available?

Reviewer #1: No

5. Is the manuscript presented in an intelligible fashion and written in standard English?

Reviewer #1: Yes

6. Review Comments to the Author

Reviewer #1: The authors extended the paper and addressed all the recommended comments and this version is well improved. This paper can considered for publication in this journal.

7. PLOS authors have the option to publish the peer review history of their article (what does this mean?). If published, this will include your full peer review and any attached files.

Reviewer #1: No

---

## [Editor Report · Acceptance letter]

12 May 2022

PONE-D-21-39391R1

Discriminatory Gleason grade group signatures of prostate cancer: An application of machine learning methods

Dear Dr. Mokoatle:

I'm pleased to inform you that your manuscript has been deemed suitable for publication in PLOS ONE. Congratulations! Your manuscript is now with our production department.

Kind regards,

on behalf of

Dr. Sathishkumar V E

Academic Editor

PLOS ONE